# Effect of Vertical High Magnetic Field on the Morphology of Solid-Liquid Interface during the Directional Solidification of Zn-2wt.%Bi Immiscible Alloy

**Bangfei Zhou** [1,†], **Xianghui Guo** [1,†], **Wenhao Lin** [1], **Ying Liu** [1], **Yifeng Guo** [1], **Tianxiang Zheng** [1,*], **Yunbo Zhong** [1,*], **Hui Wang** [2] **and Qiuliang Wang** [2]

[1] State Key Laboratory of Advanced Special Steel & Shanghai Key Laboratory of Advanced Ferrometallurgy & School of Materials Science and Engineering, Shanghai University, Shanghai 200444, China; zhoubf@outlook.com (B.Z.); gxh9708@163.com (X.G.); wenhaol@shu.edu.cn (W.L.); 18081780701@163.com (Y.L.); yfguo@shu.edu.cn (Y.G.)

[2] Institute of Electrical Engineering, Chinese Academy of Sciences Beijing, No. 6 Beiertiao, Zhongguancun, Beijing 100190, China; huiwang@mail.iee.ac.cn (H.W.); qiuliang@mail.iee.ac.cn (Q.W.)

**\*** Correspondence: ztx@shu.edu.cn (T.Z.); yunboz@staff.shu.edu.cn (Y.Z.)

**†** These authors contributed equally to this work.

**Abstract:** The morphology of the solid-liquid (S-L) interface is crucial for the directionally solidified microstructures of various alloys. This paper investigates the effect of vertical high magnetic field (VHMF) on the morphology evolution of the S-L interface and the solidified microstructure during the directional solidification of Zn-2wt.%Bi immiscible alloy. The results indicate that the morphology of the S-L interface is highly dependent on the VHMF, resulting in various solidified microstructures. When the growth rate was 1 μm/s, the aligned droplets were formed directly at the disturbed S-L interface under a 1 T VHMF. However, the stability of the S-L interface was improved to form a stable Bi-rich fiber under a 5 T VHMF. When the growth rate was 5 μm/s, the S-L interface was changed from cellular to dendritic to cellular again with increasing magnetic flux density. A theory regarding constitutional supercooling and efficient solute diffusion has been proposed to explain the S-L interface transition under the VHMF. The difference in the effective diffusion capacity of the solute originates from the thermoelectric magnetic effect and the magneto-hydrodynamic damping effect. The present work may initiate a new method to transform the solidified microstructures of immiscible alloys via an applied magnetic field during directional solidification.

**Keywords:** high magnetic field; immiscible alloy; solid-liquid interface; constitutional supercooling

## 1. Introduction

Directional solidification is one of the most widely methods for the production of high-quality components due to the precise control of microstructure [1–5]. The formation of cellular or dendritic interfaces is affected by micro-segregation and thermal profiles ahead of the interface [6–8]. These structures have an effect on the mechanical properties of the final product. Hence, the instability of a planar interface and its evolution from the planar to the cellular or dendritic interface have received particular attention from metallurgists [9–13]. At present, a large number of researches on the evolution of the solid-liquid interface are mainly aimed at the eutectic and peritectic systems, and there are few reports on the immiscible system.

Immiscible alloys are known for having a miscibility gap in their phase diagrams [14]. Moreover, it is easy to induce gravity segregation during the solidification process. However, many immiscible alloys with homogeneous structure or fibrous arrangement of the second ($L_2$) phases are regarded as useful materials for superconductors [15], wear-resistance [16], electrical contact materials [17], and so on. Directionally solidified immiscible alloys to form a desirable structure with the fibrous or dispersion of the $L_2$ phase are

considered one of the methods to develop immiscible alloy materials [14]. However, a transition from the fibrous structure to the strings of the droplet structure always occurs [18]. Several mechanisms of the structural transitions have been proposed [19–22]. These include the instability of the S-L interface due to constitutional supercooling. Therefore, it is important to recognize the morphology evolution of the solid-liquid (S-L) interface of immiscible alloys to control their solidification structures.

Researchers have explored the transitions of the S-L interface of directionally solidified immiscible alloys, but the viewpoints are difficult to unify. The structural transitions were seen early on through the observation of directional solidification of the transparent organic immiscible system [23]. Later, X-ray imaging techniques were also used to view the effect of liquid-liquid decomposition on interfacial morphology and solidification structure [24]. However, due to the limitation of the resolution, no particularly useful information is obtained. Recently, studies performed under synchrotron radiation have demonstrated the existence of both diffusive and convective mass transfer modes during the directional solidification of immiscible alloys [25]. Therefore, in order to obtain more information, researchers believe that convection ahead of the S-L interface needs to be controlled by using the magnetic field or microgravity [26].

Unlike the suppression effect of microgravity on convection, the control of convection by the magnetic field is paradoxical [27]. The static magnetic field has two main primary effects. One is the magneto-hydrodynamic damping (MHD) [28], which damps the melt flow. The other one is the thermoelectric magnetic convection (TEMC) [29], which induces particular melt convection. The morphological evolution of the S-L interface during the directional solidification of eutectic and peritectic crystals has been studied extensively under a magnetic field [30,31]. The results show that the natural convection vanishes whereas the TEMC firstly increases and then decreases with increasing magnetic flux density (MFD). Under a high MFD, the MHD effect dominated the control of melt convection, which is suppressed. In our previous work [32], it was found that the S-L interface transforms from the planar to the cellular and then to the dendritic with the increase of growth rate in the directionally solidified Zn-Bi immiscible alloy in the absence of a magnetic field. Moreover, a 5 T vertical high magnetic field (VHMF) suppresses convection, which results in the instability of the S-L interface.

In this study, we perform the directional solidification experiments at two growth rates for Zn-2wt.%Bi immiscible alloys under the VHMF with various MFDs. The relationship between the morphology of the S-L interface and the primary liquid phases is discussed. The relationship between the morphology of the S-L interface and the evolution of the solidified microstructure is established. Factors that contribute to the evolution of the morphology of the S-L interface are discussed. The multiple-control capabilities of solute transportation by VHMF may provide new insights on designing the expected immiscible alloys.

## 2. Experiments

The experimental procedures have been presented in detail in earlier work [32]. Zn-2wt.%Bi immiscible alloy was prepared by induction melting the mixed pure Zn and Bi (99.99%) particles. The rods with 4 mm in diameter and 120 mm in length were obtained through suction casting. Then, the rod was polished and subsequently cleaned to remove the surface oxide. After that, it was placed in a high-purity corundum tube for directional solidification. The experimental apparatus consists of a superconductor magnet, a typical Bridgman–Stockbarger type furnace, a temperature controller, and a pulling velocity control device. The superconducting magnet can produce a VHMF with the MFD ranging from 0 T to 5 T. The temperature in the furnace was controlled by using a K-type thermocouple with a precision of $\pm0.1$ K. The temperature gradient of the heating furnace is 160 °C/cm. The sample in the corundum crucible was remelted to 850 °C in a high-purity argon atmosphere at a heating rate of 10 °C/s. After holding for 90 min, it was subsequently withdrawn into the liquid melt (Ga-In-Sn) under various MFDs with growth rates of 1 μm/s and 5 μm/s, respectively. After a stable growth of 75 mm, the sample was

rapidly quenched (cooling rate was about 100 °C/s) in the liquid melt to preserve the S-L interface morphology. The other samples were prepared by following the above procedures. The metallography of each sample was cut parallel and perpendicular to the growth direction by wire-cut electrical discharge machining. The cut samples were mounted with epoxy resin. The mounting step was performed, followed by metallographic sanding with a sequence of sandpaper of 400 mesh, 800 mesh, 1200 mesh, and 2000 mesh. After that, the metallographic polishing was performed until a completely flat and mirror-like surface was presented. Finally, ultrasound cleaning was used to remove the contaminant. The microstructures were characterized using the backscattered electron imaging (BSE) function of a scanning electron microscope (SEM, VEGA3 SBH-Easy probe).

## 3. Results

Figure 1 shows the S-L interface morphologies and the microstructures of the directionally solidified Zn-2wt.%Bi immiscible alloy ingots at a growth rate of 1 μm/s under the various MFDs. The S-L interface firstly shows a trend of tilting and then flattening with the increase of MFD. In the absence of the VHMF (Figure 1(a1–a3)), the S-L interface is planar. Moreover, the solidified microstructure is fibrous Bi-rich phases, which eventually undergo perturbation and pinch-off during the subsequent solidification process [32]. Under a 0.1 T VHMF, the S-L interface gradually becomes unstable and the liquid Bi-rich fibers (BRFs) come to be refined, as shown in Figure 1(b1–b3). When the MFD increases to 0.5 T, the S-L interface is cellular (Figure 1(c2)), but the precipitated liquid Bi-rich phases are still fibrous (Figure 1(c3)). However, the morphology of the primary liquid Bi-rich phase changes dramatically under the 1 T VHMF. Instead of fibers, they are replaced by the Bi-rich droplet (BRD) strings. Further increasing the MFD to 5 T, the S-L interface reverts to planar and the liquid BRFs can be easily obtained. Notably, the shapes of the BRDs ahead of the S-L interface are divided into two types according to their size and arrangement. The first type is formed before quenching. The droplets grow generally larger and are independently suspended ahead of the S-L interface. Conversely, the droplets are generally smaller for the second type. They continuously distribute as the shorter BRD strings and are formed during the quenching process. Similar small BRD strings have also been obtained during the bulk quenching and melt spinning processes [33,34]. In the absence of the VHMF and with a 5 T VHMF, BRDs ahead of the S-L interface are mainly the second type. For the residual samples, BRDs belong to the first type.

Figure 2 shows morphologies of the S-L interfaces and microstructures of ingots directionally solidified at a growth rate of 5 μm/s under various MFDs. Under the effect of the VHMF, the evolution of the S-L interface is consistent with that in the case of 1 μm/s. In Figure 2(a2–f2), the S-L interface transforms from the cellular to the dendritic and then reverts to cellular with increasing MFD. However, the shapes of the primary BRDs, which are small fibers, have not been changed at the S-L interface. During the subsequent solidification process, the liquid BRFs break into the BRD strings as shown in Figure 2(a3–f3). It is known that the monotectic reaction results in a solid phase and a liquid phase, respectively, in the immiscible alloy system. This is different from the eutectic reaction in which two solid phases are formed [32]. For the hypermonotectic system, because the primary phase is a liquid, it cannot be formed as a dendrite. Instead, liquid droplets, which are the second type, are usually formed.

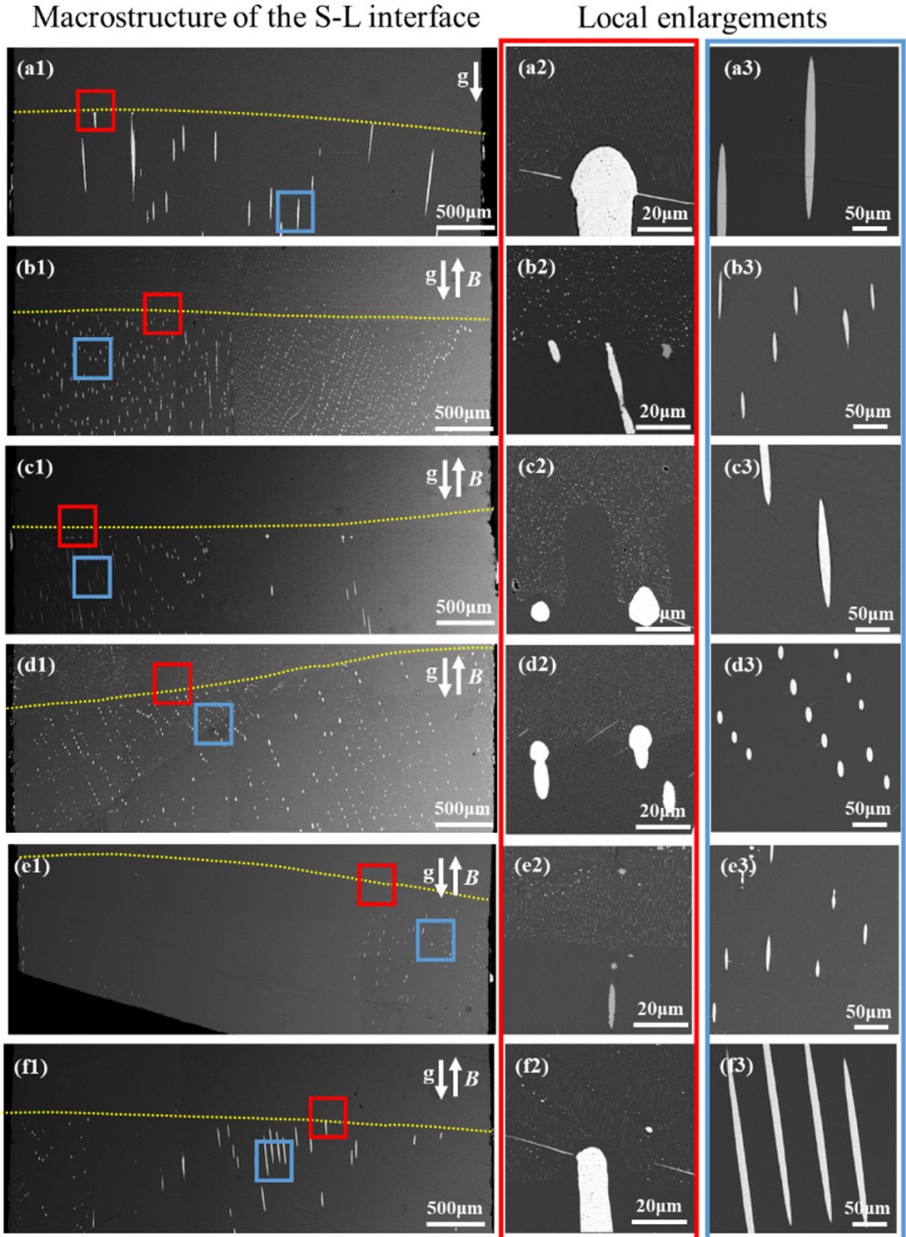

**Figure 1.** The morphology of the S-L interfaces and microstructures of ingots directionally solidified at a growth rate of 1 μm/s under various MFDs. (**a1**–**a3**) 0 T; (**b1**–**b3**) 0.1 T; (**c1**–**c3**) 0.5 T; (**d1**–**d3**) 1 T; (**e1**–**e3**) 3 T, (**f1**–**f3**) 5 T. The yellow dotted lines are used to show the S-L interfaces.

Figure 3 shows the fiber spacings on the transverse sections of ingots directionally solidified under various MFDs at growth rates of 1 μm/s and 5 μm/s, respectively. The trends of fiber spacings vary with the MFD are exactly opposite for the cases. For the growth rate of 1 μm/s, the fiber spacing decreases firstly and then increases with the increase of MFD. Under the 0.5 T VHMF, the fiber spacing reaches a minimum value of 22.5 μm, and the fiber spacings are 85 μm and 35 μm in the cases of 0 T and 5 T, respectively. In contrast, for the growth rate of 5 μm/s, the fiber spacing increases firstly and then decreases with the increase of MFD. The fiber spacing reaches a maximum value of 12.5 μm in the case of 1 T.

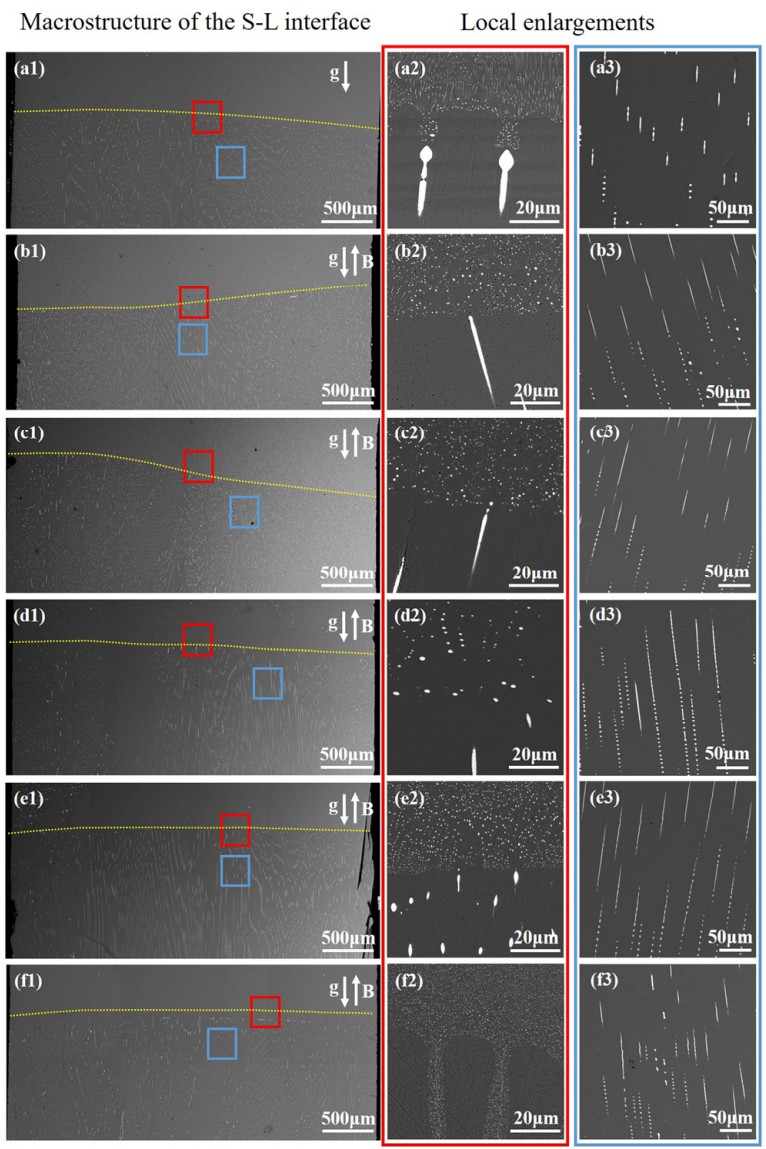

**Figure 2.** The morphology of the S-L interfaces and microstructures of ingots directionally solidified at a growth rate of 5 μm/s under various MFDs. (**a1–a3**) 0 T; (**b1–b3**) 0.1 T; (**c1–c3**) 0.5 T; (**d1–d3**) 1 T; (**e1–e3**) 3 T; (**f1–f3**) 5 T. The yellow dotted lines are used to show the S-L interfaces.

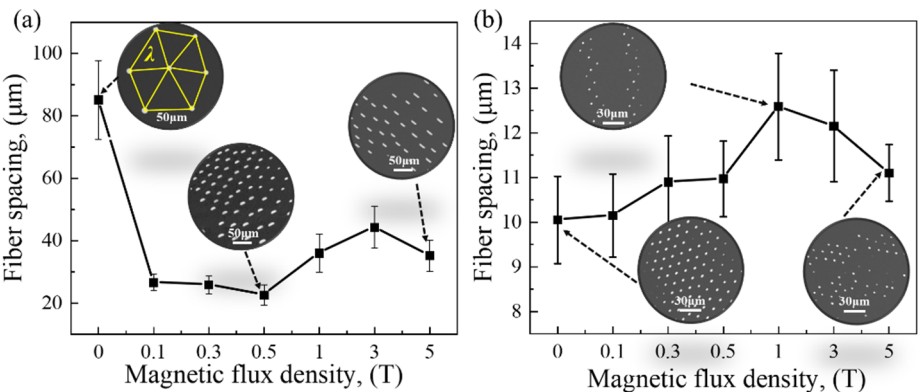

**Figure 3.** The fiber spacings (λ) on the transverse sections of ingots directionally solidified under various MFDs at growth rates of 1 μm/s (**a**) and 5 μm/s (**b**), respectively.

Figure 4 shows the microstructures and the particle size distributions of the Bi-rich particles (BRPs) ahead of the S-L interface in the ingot directionally solidified at the growth rate of 5 μm/s under different MFDs. Obviously, it is found that the size of BRP increases firstly and then decreases with the increase of MFD. When the MFD is 1 T, the size of the BRP ahead of the S-L interface reaches the maximum value.

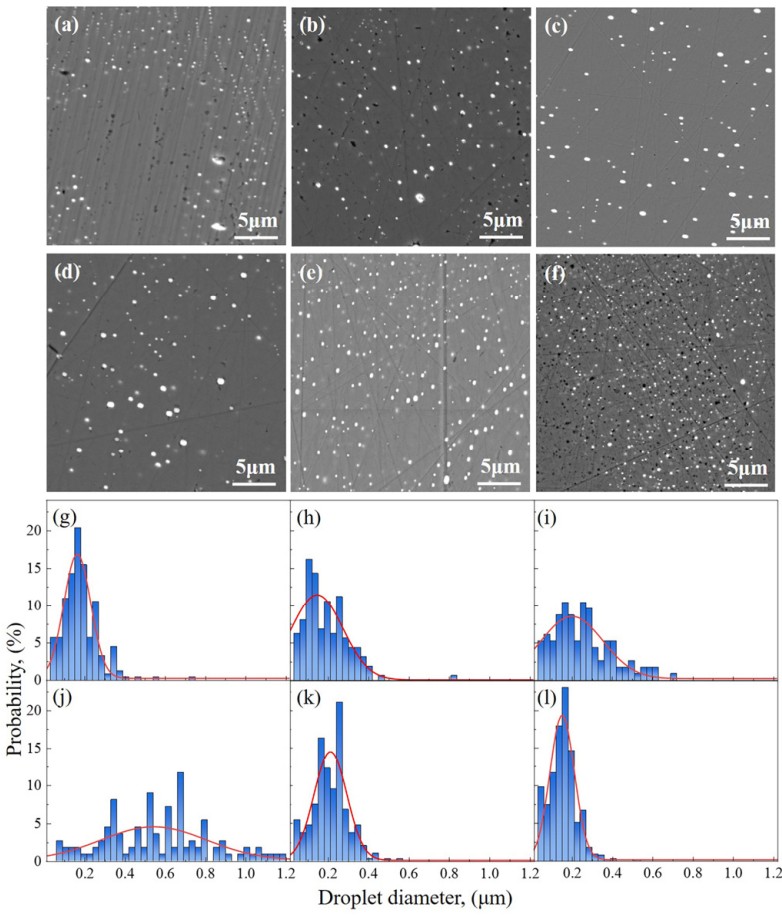

**Figure 4.** The morphologies and particle size distributions of the Bi-rich phases ahead of the S-L interface in the case of the growth rate of 5 μm/s under various MFDs. (**a,g**) 0 T; (**b,h**) 0.1 T; (**c,i**) 0.5 T; (**d,j**) 1 T; (**e,k**) 3 T; (**f,l**) 5 T. (**a–f**) The morphologies of Bi-rich droplets ahead of the S-L interface under various MFDs. (**g–l**) The size distributions of Bi-rich droplets ahead of the S-L interface under various MFDs.

## 4. Discussions

### 4.1. Morphological Evolution of the S-L Interface under Various MFDs

In the present study, for the two growth rates, the morphologies of the S-L interfaces are severely changed firstly and then revert to that in the case of no VHMF with the increase of the MFD. The critical transition occurs when the MFD is about 1 T. The schematic diagrams in Figure 5 compares and summarizes the morphological evolution of the S-L interface in the ingots directionally solidified at two growth rates under various MFDs. In fact, in our previous study [32], we found that the S-L interface transformed from the planar to the cellular and then to the dendritic with the increase of growth rate in the absence of a VHMF. The morphological evolution of the S-L interface is usually explained by the constitutional supercooling [6,8], which can be expressed as:

$$\frac{G}{V} = \frac{mC_0}{D}\frac{1-k}{k} \tag{1}$$

where $G$ is the temperature gradient, $V$ is the growth rate, $m$ is the slope of the bimodal line in the phase diagram, $C_0$ is the alloy composition, and $k$ is the partition coefficient. Therefore, the morphological evolution of the S-L interface is dependent on the growth rate, the temperature gradient, and the solute concentration ahead of the S-L interface. For a low growth rate (Figure 5(a1–a5)), a weak VHMF directly disrupts the fiber-coupled growth mode. Especially when the MFD is 1 T, the BRFs are broken into the BRD strings at the S-L interface which is formed mainly due to the constitutional supercooling [19]. When the MFD increases to 5 T, the Bi-rich phases revert to the fibers. This morphological evolution of the S-L interface leads to the fiber spacing decreasing firstly and then increasing with the increase of the MFD. However, for a relatively high growth rate (Figure 5(b1–b5)), the morphology of the S-L interface does not affect the fibrous growth of the Bi-rich phases under each MFD. Here, it should be noted that the solutes (Bi element) in the BRFs come from not only the diffusion mass transfer of the monotectic reaction, but also the convective mass transfer caused by the liquid-liquid decomposition. The strength of the convective mass transfer is mainly dependent on the MFD, and it varies under various MFDs. The above results indicate that the melt convection is enhanced by a 1 T VHMF.

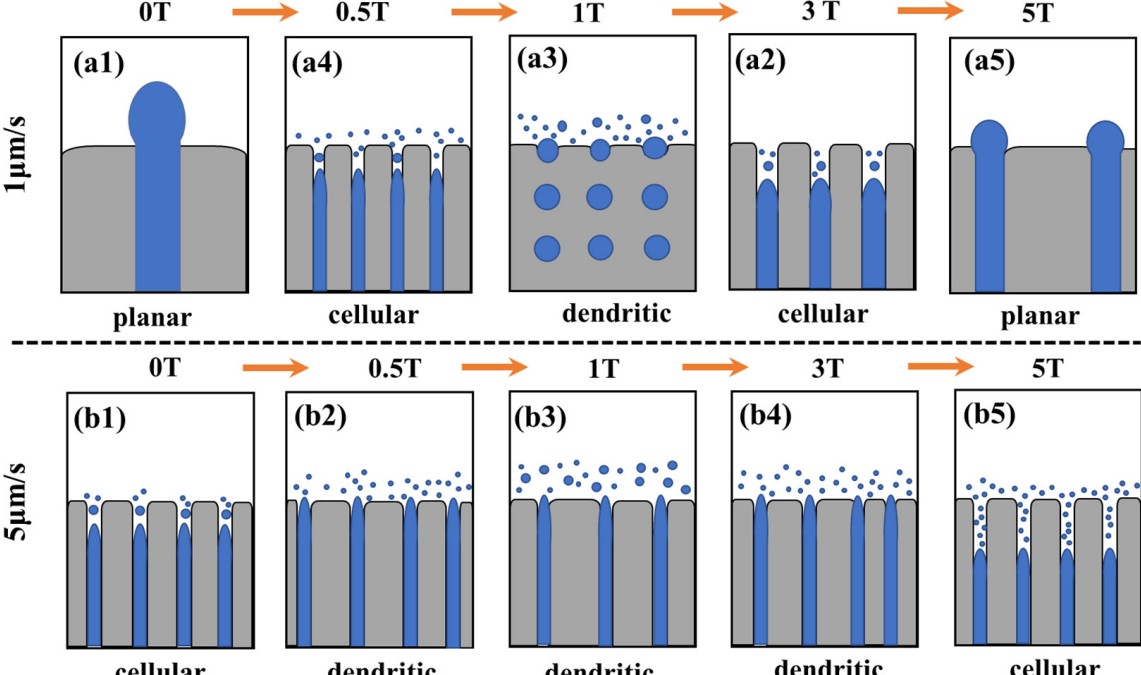

**Figure 5.** Schematic diagrams of the morphological evolution of the S-L interface in the ingots directionally solidified at growth rates of 1 μm/s and 5 μm/s, respectively, under various MFDs. (**a1–a5**) 1 μm/s; (**b1–b5**) 5 μm/s. The Bi-rich phases are colored in blue, and the Zn- rich matrixes are colored in grey.

### 4.2. Dynamic Mechanism of Morphological Evolution of the S-L Interface

The difference in the particle size distribution of the BRPs ahead of the S-L interface under various MFDs essentially reflects the change of the melt convection. Controlling the melt convection through the VHMF typically involves TEMC and MHD effects [27]. The effect of the VHMF on the melt convection can be quantified by the Hartmann number (Ha), which is defined as below:

$$\mathrm{Ha} = BL(\sigma/\rho v)^{1/2} \tag{2}$$

where $B$ denotes the MFD, $L$ is the typical length scale of the Bi-rich phase, while $\sigma$, $\rho$, and $v$ are the electrical conductivity, the density, and the kinematic viscosity of Zn-rich melt, respectively. Generally, TEMC reaches a maximum value when Ha is approximately 20 [29],

and when Ha is much greater than 1 (Ha >> 1) the convection is completely suppressed [35]. The analytical expression of TEMC was proposed by Li et al. [27]:

The case of weak VHMF

$$u_1 \approx \frac{\sigma S G B L^2}{\rho v} \tag{3}$$

The case of strong VHMF:

$$u_2 \approx \frac{SG}{B} \tag{4}$$

where $S$ is thermoelectric power. The calculated results are shown in Figure 6. It is seen that the melt velocity reaches the maximum value when the MFD is 0.8 T. The corresponding Ha is about 40. Various experimental results also show that the extreme value is reached around 0.5 T–1 T [27,30,36]. When further increasing the MFDs to 3 T and 5 T, Ha numbers are calculated to be 158 and 263, respectively. The melt convection is gradually suppressed. The numerical calculations are in good agreement with the experimental results. Thus, based on the scale of the melt velocity, the intensities of melt convection can be classified as no convection (NC), weak convection (WC), and strong convection (SC). In the absence of the VHMF and in the presence of the VHMF with a high MFD, the melt convection ahead of the S-L interface, which is regarded as NC, is relatively weak. When the MFD is around 0.8 T, the melt convection, which is regarded as SC, is the strongest. In the residual cases, the intensity of melt convection is considered as WC.

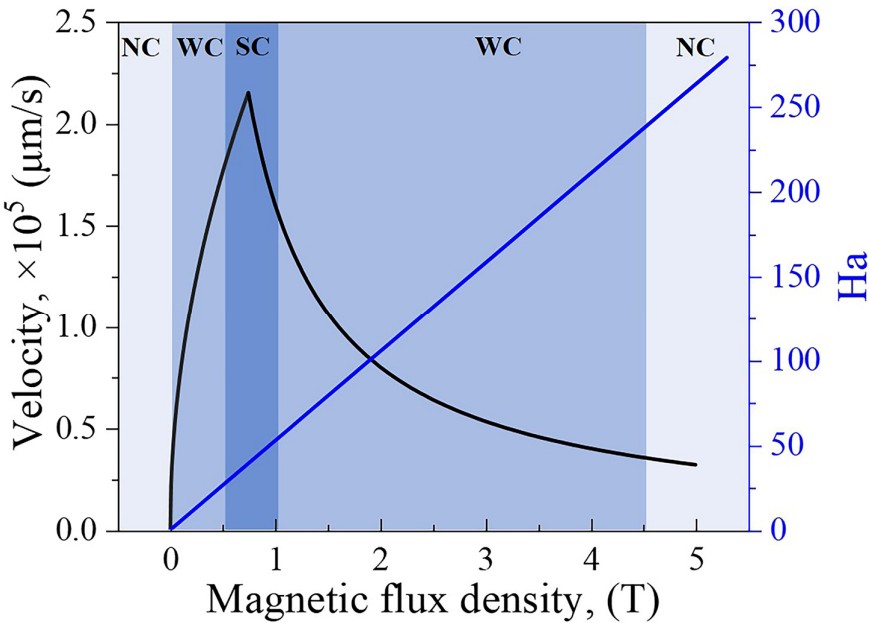

**Figure 6.** The curves of melt velocity caused by the TEMC and Ha numbers under various MFDs. NC, WC, and SC represent three different intensities of melt convection, respectively.

As we know, for a given temperature gradient, as the growth rate increases, the S-L interface will evolute as "planar → cellular → columnar-dendrite → equiaxed-dendrite". Constitutional supercooling is the fundamental theory to explain this morphological transition. For an immiscible alloy whose composition is located in the miscibility gap, its solute partition coefficient is greater than 1 ($k > 1$). During growth, the solute partition balance leads to a depletion of solutes in the liquid melt ahead of the S-L interface. Thus, a solute diffusion boundary layer ($\delta$) ahead of the S-L interface is established. The composition of the liquid ahead of the diffusion boundary layer varies from $C_0/k$ to $C_0$ with the increase of distance from the S-L interface. In a steady growth state with a low growth rate, the actual temperature of the S-L interface is equal to that of the solidus and the actual growth

velocity of the S-L interface is equal to the pulling velocity. The steady-state solute profile in the melt $C_t(x)$ is then given by Tiller [6]:

$$C_x = C_0\left\{1 + \frac{1-k}{k}exp\left(-\frac{Vx}{D}\right)\right\} \tag{5}$$

where $x$ is the distance from the S-L interface. Notably, this theory assumes that melt convection is negligible. According to Tiller's analysis, if the growth rate increases, the exponential must become steeper [6]. Therefore, an increase in the growth rate leads to a strong increase of the solute concentration gradient, as shown in Figure 7a. If only constitutional supercooling is considered, the equilibrium temperature for any point in front of the S-L interface is given by [6]:

$$T_x = T_0 - mC_0\frac{1-k}{k}exp\left(-\frac{Vx}{D}\right) \tag{6}$$

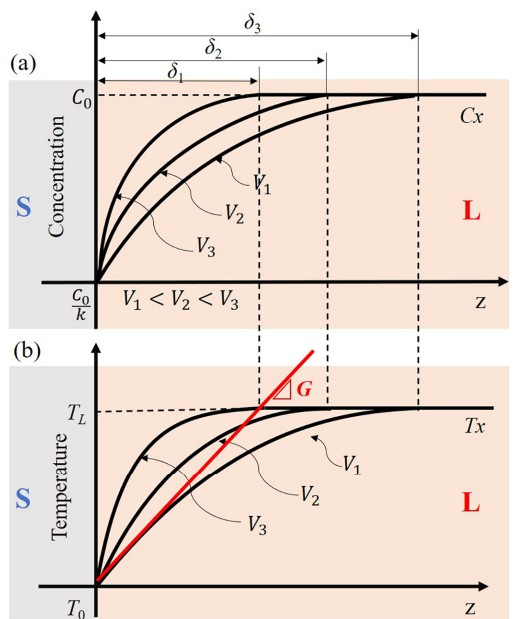 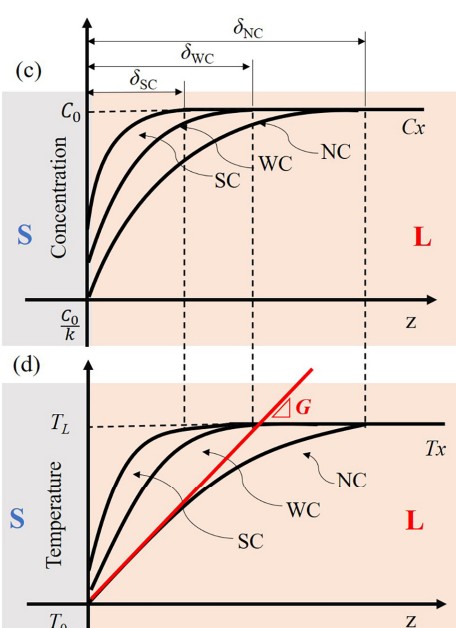

**Figure 7.** Schematic illustrations of the constitutional supercooling ahead of the S-L interface under various conditions. (**a**) Solute concentration curves under various growth rates; (**b**) Temperature curves under various growth rates; (**c**) Solute concentration curves under various intensities of melt convection; (**d**) Temperature curves under various intensities of melt convection. $\delta_1$, $\delta_2$ and $\delta_3$ denote different solute boundary layers under various growth rates, respectively. $\delta_{NC}$, $\delta_{WC}$ and $\delta_{SC}$ denote different solute boundary layers under various intensities of melt convection, respectively.

By comparing the actual temperature with the theoretical temperature, the thickness of the constitutionally supercooled zone can be plotted as shown in Figure 7b. This means that the greater the growth rate is, the easier it is to cause constitutional supercooling. Therefore, the increase of the constitutional supercooling caused by the increase of the solidification rate is the fundamental reason for the morphological transition of the S-L interface and solidification microstructure. This theory was recently confirmed by indirect observation experiments [37]. This explains why the S-L interfaces are planar and cellular when the growth rates are 1 μm/s and 5 μm/s, respectively.

In the presence of the VHMF, the magneto hydrodynamic effect has to be considered. So, the situation ahead of the S-L interface is much more complicated [26]. Comparative experiments under microgravity and normal gravity show that the reduction of solute ahead of the S-L interface induced by the melt convection is responsible for the transition from dendrite to cellular [38]. Melt convection accelerates the transport of solutes ahead of

the S-L interface and reduces the thickness of the solute diffusion boundary layer ($\delta$) [13]. For alloys with $k > 1$, convection accelerates the transfer of solutes between the solute diffusion boundary layer and the region far away from this layer. Therefore, the different convection intensities will result in different solute concentration gradients. The effect of melt convection on the solute distribution and the constitutional supercooling under various MFDs are shown in Figure 7c,d, respectively. From the previous calculations (Figure 6), it can be seen that the VHMF directly affects the intensity of the melt convection ahead of the S-L interface. In the case of NC, the solute diffusion boundary layer ($\delta_{NC}$) is thick, and the solute concentration gradient is relatively low. Then, the constitutional supercooling can not be formed ahead of the S-L interface resulting in a planar S-L interface. In case of the WC, convection promotes the solute migration to the interface, leading to a shorter distance of the solute diffusion boundary layer ($\delta_{WC}$) and a higher solute concentration gradient. Then, the constitutional supercooling increases and results in a cellular interface. When the MFD reaches about 1 T, the melt convection is the strongest due to the TEMC, which can be regarded as SC. At this time, the solute diffusion boundary layer ($\delta_{SC}$) thickness reaches the minimum and solute concentration gradient reaches the maximum, resulting in a dendritic interface. When the MFD is greater than 1 T, the melt convection is gradually weakened according to Figure 6. As a result, constitutional supercooling also continues to be decreased. Then, a planar S-L interface is formed. The profound changes in the morphology of the S-L interface result in the transformation of the solidified microstructures under various MFDs at a low growth rate of 1 μm/s. In particular, a strong convection destroys the stability of the planar interface which leads to the increase of the groove of cell. Then, the spacing of the fibers gradually decrease when the MFD increases to 1 T. When the MFD further increases from 1 T to 5 T, the intensity of TEMC is gradually reduced. This makes the S-L interface be stable again. Then, the fiber spacing increases as a result. However, it should be noted that, under a BHMF, the fiber spacing must be smaller than that under 0 T. This is attributed to the reduction of the effective diffusion capacity of the solute.

The above analysis model is also suitable for the situation at the high growth rate. For the growth rate of 5 μm/s, the S-L interface is cellular in the absence of a VHMF. Since the MFD increases to 1 T, the melt convection gradually increases and results in a decrease in the solute boundary layer and an increase in the solute concentration gradient. When the MFD continues to increase, the thickness of the solute boundary layer increases and the solute concentration gradient decreases. Therefore, the morphology of the S-L interface transits from cellular to dendritic and then to cellular with the increase of the MFDs. The directionally solidified microstructure at the high growth rates are fibrous, which have not been changed with the increase of the MFD. This is different from the situation at the low growth rates. According to Ratke's simulation [39], melt convection increases the fiber spacing, if the solidified structure maintains the fiber structure. Therefore, with the growth rate of 5 μm/s, the fiber spacing increases firstly and then decreases with the increase of the MFD. It should be noted that this simplified analysis has ignored the effect of melt convection on the heat transfer. Melt convection will lead to a uniform melt temperature ahead of the S-L interface and thus reduce the temperature gradient. However, this does not change the theory that melt convection increases the constitutional supercooling. In addition, it should be emphasized that many factors have effects on the morphology of the S-L interface, such as surface energy and melting entropy. In this study, the simplified analysis can better explain the morphological evolution of the S-L interface during the directional solidification of Zn-2wt.%Bi immiscible alloys under various MFDs. However, the actual quantitative determination still needs support from more experimental data in the future.

## 5. Conclusions

In this study, the effects of the VHMF on the morphology of the S-L interface and microstructures of the directionally solidified Zn-2wt.%Bi immiscible alloys are studied.

Several transitions are found regarding the morphological evolution of the S-L interface and microstructures under various MFDs. For a low growth rate of 1 μm/s, the S-L interfaces are changed as "planar → cellular → dendritic → cellular→ planar" with the increase of the MFD. Moreover, the corresponding microstructures are changed as "fiber → BRD string → fiber". For a high growth rate of 5 μm/s, the S-L interfaces are changed as "cellular → dendritic → cellular" with the increase of the MFD. However, all the Bi-rich phases display as fiber. The effects of TEMC and MHD on the melt convection under the VHMF have been quantified. The intensity of melt convection ahead of the S-L interface increases firstly and then decreases with the increase of the MFD. The increase of both the growth rate and melt convection leads to the reduction of the thickness of the solute boundary layer and the increase of the solute concentration gradient ahead of the S-L interface. Moreover, these changes lead to an increase in the constitutional supercooling, which interprets the morphological evolution of the S-L interface. This study provides a new understanding of morphological evolutions of the directionally solidified microstructures in immiscible alloys. Besides, this study also opens new possibilities for using a high magnetic field to process the immiscible alloy with an expected microstructure.

**Author Contributions:** Conceptualization, B.Z. and Y.Z.; methodology, B.Z. and X.G.; formal analysis, B.Z., X.G., Y.Z. and T.Z.; data curation, B.Z. and Y.Z.; writing—original draft preparation, B.Z. and Y.Z.; writing—review and editing, W.L., Y.L., Y.Z. and T.Z.; project administration, Y.G. and Y.Z.; Resources, H.W. and Q.W. funding acquisition. All authors have read and agreed to the published version of the manuscript.

**Funding:** The authors gratefully acknowledge the National Key Research and Development Program of China (Grant 2018YFF0109404), the financial support of the National Natural Science Foundation of China (Grant U1732276), Natural Science Foundation of Shanghai (Grant 21ZR1424400), the China Postdoctoral Science Foundation (Grant 2021M692020), Open Project of State Key Laboratory of Advanced Special Steel Shanghai Key Laboratory of Advanced Ferrometallurgy, Shanghai University (Grant SKLASS 2021-Z06), Changjiang Scholars Program of China, China Association for Science and Technology Young Talent Support Project.

**Institutional Review Board Statement:** Not applicable.

**Informed Consent Statement:** Not applicable.

**Data Availability Statement:** Data sharing is not applicable.

**Conflicts of Interest:** The authors declare that they have no known competing financial interests or personal relationships that could have appeared to influence the work reported in this paper.

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
