# Peer review of "Effect of Vertical High Magnetic Field on the Morphology of Solid-Liquid Interface during the Directional Solidification of Zn-2wt.%Bi Immiscible Alloy"

_metals, doi:10.3390/met12050875_

Round 1
Reviewer 1 Report
Comments:
- the article contains many editorial errors, especially the references section should be noted, some items are incorrectly given (fonts, missing initials, etc.), including: 1, 36, 37, 38, 39;
- on page 8 the font should be corrected from the sentence: "where S is thermoelectric power"
- authors should consider providing more recent literature (15 out of 39 of the last 10 years);
- provide more details about the experiment, heating times, etc.;
- the sentence "After a stable growth for 75 mm, the sample was rapidly quenched in the liquid melt to preserve the S-L interface morphology" is unclear. We should better describe "rapidly quenched", specifically what was the cooling rate (even estimated), in which atmosphere the samples were cooled?
- how were the samples cut, how was their surface prepared for testing?
- on what basis did the authors choose the MFD values, shouldn't there be an intermediate value between 1 T and 5 T? For the velocity of 5 μm/s, the results for 3 T were given. Using the data from Fig. 6, other MFD values ​​could be selected for a better comparison of the calculations and the experiment;
- Sentences "For the growth rate of 1 μm/s, the fiber spacing decreases firstly and then increases with the in-crease of MFD" and "In contrast, for the growth rate of 5 μm/s, the fiber spacing increases firstly and then decreases with the increase of MFD ”are quite significant. This effect should be better explained. Again, the question, maybe you should provide data for more MFDs? Figure 5 shows very well the difference of structures created during different growth rates, but the comparison with the same MFD values ​​is more convincing.
Overall, I think the article is quite interesting and brings new knowledge. It is true that the authors only took pictures of the microstructure and calculations, but it was carried out in a thoughtful way. I believe that the topic is developmental and the authors will certainly continue their research. Subject to changes and clarification of doubts, the article may be published in Metals.
Reviewer 2 Report
I have carefully reviewed the manuscript “Effect of vertical high magnetic field on the morphology of solid-liquid interface during the directional solidification of Zn- 2wt%Bi immiscible alloy” and would like to recommend foe the publication of the manuscript with some minor changes.
- The abstract of the manuscript should be cut short and must indicate the novelty of the work.
- The future outlook of the “solid-liquid interface” can be added in the introduction section. The solid-liquid interface is currently widely used in 2D materials for large area synthesis and many other materials syntheses as well. Few lines about that will make a good impact of the introduction section. You can refer to the article “Chemical Society Reviews 50 (7), 4684-4729”
- Please carefully proofread the Introduction and discussion section as there are few typos.
Overall, the manuscript is novel and well structured. The manuscript can be accepted after the above changes are completed.
Round 2
Reviewer 1 Report
The authors responded to my comments and revised the manuscript. I believe the article may be published in Metals.